# The EGFR-ERK/JNK-CCL20 Pathway in Scratched Keratinocytes May Underpin Koebnerization in Psoriasis Patients

**DOI:** 10.3390/ijms21020434

**Published:** 2020-01-09

**Authors:** Kazuhisa Furue, Takamichi Ito, Yuka Tanaka, Akiko Hashimoto-Hachiya, Masaki Takemura, Maho Murata, Makiko Kido-Nakahara, Gaku Tsuji, Takeshi Nakahara, Masutaka Furue

**Affiliations:** 1Department of Dermatology, Faculty of Medical Sciences, Kyushu University, Maidashi 3-1-1, Fukuoka 812-8582, Japan; ffff5113@gmail.com (K.F.); takamiti@dermatol.med.kyushu-u.ac.jp (T.I.); yukat53@med.kyushu-u.ac.jp (Y.T.); take0917@dermatol.med.kyushu-u.ac.jp (M.T.); muratama@dermatol.med.kyushu-u.ac.jp (M.M.); macky@dermatol.med.kyushu-u.ac.jp (M.K.-N.); 2Research and Clinical Center for Yusho and Dioxin, Kyushu University Hospital, Maidashi 3-1-1, Fukuoka 812-8582, Japan; ahachi@dermatol.med.kyushu-u.ac.jp (A.H.-H.); gakku@dermatol.med.kyushu-u.ac.jp (G.T.); 3Division of Skin Surface Sensing, Department of Dermatology, Faculty of Medical Sciences, Kyushu University, Maidashi 3-1-1, Fukuoka 812-8582, Japan; nakahara@dermatol.med.kyushu-u.ac.jp

**Keywords:** CCL20, IL-17A, psoriasis, scratch injury, epidermal growth factor receptor, Koebner phenomenon, ERK, JNK, p38 MAPK, keratinocyte

## Abstract

Epidermal keratinocytes represent a rich source of C-C motif chemokine 20 (CCL20) and recruit CCR6^+^ interleukin (IL)-17A–producing T cells that are known to be pathogenic for psoriasis. A previous study revealed that scratch injury on keratinocytes upregulates CCL20 production, which is implicated in the Koebner phenomenon characteristically seen in psoriasis patients. However, the molecular mechanisms leading to scratch-induced CCL20 production remain elusive. In this study, we demonstrate that scratch injury upregulates the phosphorylation of epidermal growth factor receptor (EGFR) and that the specific EGFR inhibitor PD153035 attenuates scratch-induced CCL20 upregulation in an extracellular signal-related kinase (ERK)-dependent, and to a lesser extent, a c-Jun N-terminal kinase (JNK)-dependent but p38 mitogen-activated protein kinase (MAPK)–independent manner. Immunoreactive CCL20 was visualized in the keratinocytes that lined the scratched wound. IL-17A also induced the phosphorylation of EGFR and further augmented scratch-induced CCL20 upregulation. The EGFR-ERK/JNK-CCL20 pathway in scratched keratinocytes may explain why Koebnerization is frequently seen in psoriasis patients.

## 1. Introduction

Psoriasis is an immune-mediated skin disease characterized by epidermal hyperproliferation and dermal inflammatory cell infiltrates such as dendritic cells and T cells [1]. Psoriasis has shown a diverse prevalence across populations worldwide: 2.5% in Europeans, 0.05–3% in Africans, and 0.1–0.5% in Asians [1,2]. Psoriasis markedly diminishes the quality of life, treatment adherence/satisfaction, and socioeconomic stability of afflicted patients [3,4]. Skin injury triggers or exacerbates psoriatic lesions in a condition named the Koebner phenomenon [5,6]. Although the Koebner phenomenon is observed in other skin diseases such as lichen planus and vitiligo, it is particularly associated with psoriasis [5,6]. However, the pathogenetic mechanism of Koebnerization is not clear [5,6].

A number of biologic therapies are currently approved for the management of moderate to severe psoriasis. These biologics target specific molecules in the immune system, and they have a favorable safety and efficacy profiles than the traditional systemic agents such as methotrexate and cyclosporine [7]. Therapeutic success by specific biologics points to the pathogenic role of the tumor necrosis factor-α (TNF-α) axis and the interleukin (IL)-23/IL-17A axis in psoriasis [8,9,10,11,12,13]. The gene expression of *TNF*, *IL23*, and *IL17A* is upregulated in the skin lesions of psoriasis patients [13,14,15]. Infiltration of IL-17A–producing T helper (Th17) cells is detected in the lesional skin of psoriatic patients, and certain Th17 cells are reactive to selective autoantigens [16,17,18].

The recruitment of Th17 cells into the lesion is governed by CCL20-CCR6 engagement [19,20]. The expression of CCR6 has been confirmed in other IL-17A–producing cytotoxic T cells (Tc17) [21,22], innate lymphoid cell group3 (ILC3) [23,24], and γδT cells [25,26]. CCL20 is a potent chemoattractant for CCR6^+^ T cells as well as dendritic cells [20,27,28,29]. Psoriatic lesions are associated with abundant epidermal CCL20 expression and dermal skin–homing CCR6^+^ Th17 cells [17,30,31]. Epidermal keratinocytes represent a rich source of CCL20 secretion [32]. In addition, mechanical suctioning or scratching upregulates the mRNA and protein expression of CCL20 [27,32], and keratinocytes release large amounts of CCL20 in a time- and scratch line number-dependent manner [32]. In a murine psoriasis model generated by intradermal IL-23 injection, treatment with an anti–CCL20 antibody significantly reduced the recruitment of CCR6^+^ cells and attenuated IL-23–induced psoriasiform dermatitis [33]. Getschman et al. designed a CCL20 variant, CCL20S64C, that acts as a partial agonist of CCR6 [34]. After administration, CCL20S64C competes with CCL20 and significantly attenuates IL-23–induced psoriasiform inflammation in mice [34]. These preclinical studies reinforce the crucial role of the CCL20-CCR6 axis in the pathogenesis of psoriasis.

We have previously demonstrated an upregulated production of CCL20 following scratch injury in keratinocytes and proposed a potential link to the Koebner phenomenon in psoriasis [32]. However, the subcellular mechanisms of scratch-induced CCL20 production in keratinocytes remain elusive. One of the prominent biological alterations following scratch wounding is the activation of epidermal growth factor receptor (EGFR) in epithelial cells, including keratinocytes and corneal cells [35,36]. Therefore, we hypothesized that EGFR activation induces upstream signal transduction for CCL20 production. In this study, we demonstrated that scratch-induced CCL20 production was mediated by EGFR-extracellular signal-related kinase (ERK), and to a lesser extent, by the EGFR–c-Jun N-terminal kinase (JNK) pathway in keratinocytes. IL-17A also upregulated CCL20 production via EGFR activation and further potentiated scratch-induced CCL20 production, suggesting that epidermal CCL20 production is an integral part in the pathogenesis of psoriasis and Koebnerization.

## 2. Results

### 2.1. Scratch-Induced CCL20 Expression Is Ameliorated by EGFR Inhibition

Consistent with our previous report [32], scratch injury augmented the protein production of CCL20 compared with non-scratched control human keratinocytes (Figure 1). Significant amounts of CCL20 were released from scratched keratinocytes as early as 3 h after scratch injury (Figure 1). Notably, the EGFR inhibitor PD153035 significantly inhibited scratch-induced CCL20 upregulation (Figure 2A). Moreover, PD153035 significantly decreased the baseline production of CCL20 even in non-scratched controls (Figure 2A). We next examined whether or not scratch injury phosphorylates EGFR. In accordance with previous reports [35,36], scratch injury upregulated the phosphorylation of EGFR (P-EGFR) compared with non-scratched controls, and scratch-induced P-EGFR upregulation was attenuated in the presence of PD153035 (Figure 2B). These results suggest a pivotal regulatory role of EGFR signaling in scratch-induced CCL20 production in keratinocytes.

### 2.2. Spatial Distribution of CCL20 Expression in Keratinocytes

Subsequently, we attempted to visualize the CCL20 expression using an immunofluorescence technique. Compared with IgG staining in negative controls (Figure 3A), the immunoreactive CCL20 was positively but faintly and diffusely stained in the non-scratched control keratinocytes (Figure 3B). Compared with staining in negative controls (Figure 3C), CCL20 expression was clearly enhanced in the keratinocytes residing along the scratch-edge area (Figure 3D, arrows). These findings strongly suggest that the scratch injury triggers CCL20 production in keratinocytes adjacent to the wound.

### 2.3. Scratch-Induced CCL20 Production Depends on ERK and, to a Lesser Extent, JNK, but Not the p38 Mitogen-Activated Protein Kinase (MAPK) Pathway

The variable involvement of MAPKs (ERK, JNK, and p38 MAPK) in CCL20 expression has been reported in stimuli-specific and cell type–specific manners [37,38,39,40,41]. Therefore, we examined whether scratch injury activates MAPKs in our system. As shown in Figure 4, scratch injury phosphorylated ERK, JNK, and p38 MAPK molecules compared with nonscratched controls. We then investigated whether scratch-induced CCL20 expression is affected by the ERK inhibitor U0126, the JNK inhibitor SP600125, and the p38 MAPK inhibitor SB203580 (Figure 5). Of note, the scratch-induced CCL20 expression was strongly and significantly inhibited by U0126 and partially by SP600125 (Figure 5). U0126 and SP600125 also inhibited the baseline production of CCL20 in nonscratched keratinocytes (Figure 5). SB203580 did not inhibit baseline and scratch-induced CCL20 production (Figure 5). These results indicate that scratch-induced CCL20 expression is regulated by ERK and partially by JNK activation in keratinocytes.

### 2.4. IL-17A Synergistically Enhances Scratch-Induced CCL20 Production

Because IL-17A feasibly upregulates CCL20 production in keratinocytes [42], we next checked whether IL-17A augments scratch-induced CCL20 expression. IL-17A alone upregulated CCL20 production in non-scratched control keratinocytes in a dose-dependent fashion (Figure 6A). When combined with scratch injury, IL-17A additively or synergistically enhanced scratch-induced CCL20 production (Figure 6A). Interestingly, the IL-17A–induced CCL20 production was also attenuated by the EGFR inhibitor PD153035 (Figure 6B). In accordance with this result, IL-17A alone induced the phosphorylation of EGFR and was attenuated by PD153035 in non-scratched control keratinocytes (Figure 7). IL-17A induced similar levels of EGFR activation as did scratch injury. Significant enhancement of phosphorylated EGFR (P-EGFR) was not observed in combined treatment with scratch and IL-17A compared with IL-17A monotreatment (Figure 7). These results stress the crucial role of EGFR activation in IL-17A–induced, as well as in scratch-induced CCL20 production.

## 3. Discussion

CCR6 is a representative surface marker for IL-17A–producing immune cells [19,20,21,23,28,43]. CCL20 and CCR6 engagement is critical for the recruitment of CCR6^+^ Th17 cells [27,29,44]. Although immune cells can feasibly produce CCL20 [45,46,47,48], the expression of CCL20 seems to be confined to peripheral tissues, including skin [32,42,44]. Epidermal keratinocytes constitutively produce CCL20 in culture conditions [32,42]. The lesional epidermal keratinocytes express abundant CCL20 in patients with psoriasis [30,31]. Following stimulation with IL-17A, keratinocyte production of CCL20 is markedly enhanced [42].

We have previously shown that mechanical scratching also upregulates CCL20 production in keratinocytes [32]. However, the underlying mechanisms remain elusive. In this study, we demonstrated that scratch-induced CCL20 expression is mediated by the EGFR-ERK (and, to a lesser extent, by the EGFR-JNK) pathway. The scratch injury induced the phosphorylation of EGFR in keratinocytes. The scratch-induced activation of EGFR is consistent with previous reports [35,36]. The production of CCL20 was detected as early as 3 h after scratch injury, and the blockade of EGFR activation by its specific inhibitor, PD153035, significantly reduced scratch-induced CCL20 upregulation. PD153035 also significantly decreased the baseline production of CCL20 in non-scratched control keratinocytes. These results indicate that CCL20 production is highly dependent on EGFR signaling. Notably, the immunofluorescence study revealed that CCL20 expression was markedly augmented in keratinocytes lining the scratched edge, suggesting a clear relationship between scratch injury and CCL20 upregulation.

Scratch injury activated and induced the phosphorylation of ERK1/2, JNK, and p38 MAPK in keratinocytes. However, in the present study, scratch-induced CCL20 protein upregulation was strongly blocked by an ERK inhibitor and partially by a JNK inhibitor but not by a p38 MAPK inhibitor. In hepatocellular carcinoma cells and murine colonic epithelial cells, CCL20 expression is known to depend on the ERK pathway [37,38]. TNF-α–induced CCL20 upregulation depends on ERK and p38 MAPK in bronchial epithelial cells [39]. Prolactin upregulated CCL20 expression in keratinocytes and was dependent on ERK and JNK [40]. Evidence from our own and previous studies suggests that the ERK (partially JNK) pathway plays a pivotal role in scratch/EGFR-mediated CCL20 production. Although p38 MAPK is not directly involved in scratch-induced CCL20 production, a recent study suggests that p38 MAPK may regulate the recycling of EGFR by accelerating the latter’s endocytosis [49].

Positive therapeutic responses to biologics and lesional transcriptomic analysis have highlighted the critical pathogenic role of IL-17A in psoriasis [15,32,50]. Scratch injury triggers the development of psoriatic lesions (Koebner phenomenon) in patients with psoriasis [6]. Therefore, we next examined the effects of IL-17A on scratch-induced CCL20 production. IL-17A dose-dependently increased CCL20 production in non-scratched control keratinocytes, as reported previously [42]. When they were combined with the scratch injury, we observed an additive or synergistic augmentation of CCL20 production by scratch and IL-17A. In line with a previous study that elucidated EGFR activation by IL-17A [51,52], we also demonstrated that IL-17A upregulated the phosphorylation of EGFR and that the EGFR inhibitor PD153035 attenuated IL-17A–mediated CCL20 production. These results suggest that, in patients with psoriasis, the scratch injury stimulates keratinocytes to produce CCL20, which attracts circulating Th17 cells into the scratched skin. IL-17A produced from recruited Th17 cells may further accelerate CCL20 production from the scratched epidermis and further recruit circulating Th17 cells to the injured skin, leading to a feed-forward positive loop that develops into the Koebnerized psoriatic eruption. Both scratch-induced and IL-17A–induced CCL20 expression depend on EGFR activation. Consistent with the present results, it has been demonstrated that the inhibition of EGFR by erlotinib or cetuximab successfully improves severe psoriasis [53,54,55,56].

A first-in-human study using a humanized anti–CCL20 antibody, GSK3050002, has already been reported for the treatment of psoriasis [27]. Intravenous GSK305002 inhibits the infiltration of Th17 cells, but not Th1 or Th2 cells, into the suction blisters (rich in CCL20) in treated individuals, suggesting the active and selective involvement of the CCL20/CCR6 axis in Th17 cell homing into the skin [27]. In addition, preclinical studies reveal that a humanized anti–CCR6 antibody or a small-molecule CCR6 antagonist efficiently inhibits the infiltration of immune cells and inflammatory symptoms in a murine psoriatic model [29,43]. These studies stress the significance of CCL20 as a potential therapeutic target for psoriasis. In parallel, the increased expression of CCL20 and CCR6 has been demonstrated in lichen planus which also manifests Koebnerization [57]. There are several limitations in this study. We only examined the effects of scratch injury on the monolayer keratinocyte culture system. Further studies are warranted using three-dimensional skin equivalent, ex vivo human skin, or in vivo mouse models. Fibroblasts may affect the scratch-induced, keratinocyte-derived CCL20 expression in coculture models. More extensive studies may be necessary to draw a concrete conclusion.

In conclusion, scratch-induced CCL20 production depends on the EGFR-ERK (and partially on the JNK) pathway. As has been proposed previously [58], EGFR signaling may be a significant and integral component in the pathogenesis of Koebnerization and psoriasis. Our study underscores the potential role of new therapies such as anti-CCL20 antibody and EGFR/ERK inhibitors in the management of psoriasis.

## 4. Materials and Methods

### 4.1. Reagents and Antibodies

Dimethyl sulfoxide (DMSO) was purchased from Sigma-Aldrich (St. Louis, MO, USA). CaCl_2_ (Fujifilm Wako Pure Chemical Corporation, Osaka, Japan) was dissolved in UltraPure distilled water (Invitrogen, Carlsbad, CA, USA) and added to culture medium at a final concentration of 1.6 mM. In the same way, recombinant human IL-17A (PeproTech, Rocky Hill, NJ, USA) was used at final concentrations of 0.1 and 1.0 ng/mL. An EGFR–tyrosine kinase inhibitor, PD153035 (ChemScene LLC, Monmouth Junction, NJ, USA), was dissolved in DMSO and added to culture medium at final concentrations of 300 and 600 nM. Control cultures contained comparable amounts of DMSO (0.01%) and UltraPure distilled water (1.6%). Signal-transduction inhibitors U0126 (ERK1/2 inhibitor) and SB203580 (p38 inhibitor) were purchased from Tocris Bioscience (Bristol, UK), and SP600125 (JNK inhibitor) was obtained from Abcam (Cambridge, UK). The antibodies used for immunofluorescence staining were as follows: rabbit anti–human macrophage inflammatory protein 3 alpha (MIP-3α, CCL20) polyclonal antibody, and normal rabbit polyclonal IgG (all from Abcam); and goat anti–rabbit IgG (H + L) cross-adsorbed secondary antibody with Alexa Fluor 488 dye (Thermo Fisher Scientific, Waltham, MA, USA). The antibodies (catalog numbers) used for western blotting were as follows: rabbit anti–human ERK1/2 (9102S), JNK (9252S), p38 MAPK (8690S), phospho-ERK1/2 (4370S, Thr202/Tyr204), phospho-JNK (4668S, Thr183/Tyr185), phospho-p38 MAPK (4511S, Thr180/Tyr182), EGFR (4267S), phosphor-EGFR (3777S, Tyr1068), and β-actin (4970S) monoclonal antibodies (all from Cell Signaling Technology, Danvers, MA, USA) as primary antibodies, and goat anti-rabbit IgG HRP-linked antibody (7074S) (Cell Signaling Technology) as a secondary antibody.

### 4.2. Cell Culture

Normal human epidermal keratinocytes (NHEKs) from neonatal foreskin were cultured in KBM-GOLD Keratinocyte Basal Medium supplemented with KGM-GOLD SingleQuots (all from Lonza, Basel, Switzerland) containing bovine pituitary extract, recombinant epidermal growth factor, insulin, hydrocortisone, transferrin, gentamicin sulfate–amphotericin (GA-1000) (Lonza), and epinephrine and were maintained at 37 °C in 5% CO_2_. The culture medium was replaced every day, and the cells were serially passaged at 70–80% confluence. In all experiments, cells were used at the third passage. Cells (3.5 × 10^5^ cells/well) were seeded in 6-well culture plates (Corning Inc., Corning, NY, USA), and the culture medium was replaced every day. At 100% confluence, cells were treated with a high level of CaCl_2_ (1.6 mM, high-Ca^2+^ conditions), which converted the cells from the proliferative state to the differentiating state.

### 4.3. CCL20 Secretion in an In Vitro Scratched Keratinocyte Model

The in vitro scratched keratinocyte model was performed as previously described [32]. Briefly, NHEKs at 100% confluence were incubated for 24 h under high-Ca^2+^ conditions at 37 °C in 5% CO_2_. Then, the cell sheets were scratched (10 lines) with a 250-μL Long Tip (Watson Bio Lab, Tokyo, Japan). In several assays, the scratched cell sheets were also treated with IL-17A (0.1 or 1.0 ng/mL) [42,59], PD153035 (300 or 600 nM) [60], U0126 (10 μM), SB203580 (10 μM), or SP600125 (10 μM).

### 4.4. Enzyme-Linked Immunosorbent Assay (ELISA)

Culture supernatants were collected, and the concentrations of CCL20 were measured using Quantikine Human CCL20 ELISA Kits (R&D Systems, Minneapolis, MN, USA) in accordance with the manufacturer’s instructions. Absorbance at 450 nM was measured with an iMark microplate reader (Bio-Rad Laboratories Inc., Hercules, CA, USA), and the concentrations of CCL20 were determined in each sample by comparison with a standard curve.

### 4.5. Western Blotting

Protein lysates of NHEKs were isolated from cells with 100 μL/well of lysis buffer (25 mM HEPES, 10 mM Na_4_P_2_O_7_/10 H_2_O, 100 mM NaF, 5 mM EDTA, 2 mM Na_3_VO_4_, and 1% Triton X-100), supplemented with 10 μL/well of proteinase inhibitor cocktail (Sigma-Aldrich). The lysates were then centrifuged at 14,000 rpm for 20 min, and the supernatants were used for analysis. The protein concentration of each lysate was measured with a BCA protein assay kit (Thermo Fisher Scientific). Equal amounts of protein (30 μg for EGFR, phosphor-EGFR; 20 μg for the other proteins) were mixed with 2× sample buffer (Nacalai Tesque, Inc., Kyoto, Japan), boiled at 95 °C for 5 min, loaded onto Bolt 4–12% Bis-Tris Plus Gels (Invitrogen), and electrophoresed at 200 V and 160 mA for 22 min. The proteins were then transferred to a polyvinylidene fluoride (PVDF) membrane (Invitrogen) using the Power Blotter XL System (Invitrogen). Membranes were blocked with 2% bovine serum albumin (BSA; Sigma-Aldrich) in 0.1% tris buffered saline with tween (TBS-T). Membranes were probed with primary antibodies overnight at 4 °C. After incubation with anti–rabbit IgG HRP-linked secondary antibody at room temperature for 1 h, protein bands were visualized with SuperSignal West Pico Chemiluminescent Substrate (Thermo Fisher Scientific) using the ChemiDoc Touch Imaging System (Bio-Rad Laboratories Inc.).

### 4.6. Immunofluorescence Analysis

Immunofluorescence analysis was performed as reported previously [61]. The cells were cultured in 4-well slide chambers (1.5 × 10^5^ cells/well) (Lab-Tek, Rochester, NY, USA) in accordance with the culture methods described herein. Then, the cell sheets were scratched and incubated for 6 h at 37 °C in 5% CO_2_. The cell sheets were washed with PBS 3 times for 5 min each and fixed in cold acetone for 10 min at room temperature. The cell sheets were blocked with 10% BSA (Roche Diagnostics, Basel, Switzerland) and incubated with rabbit anti-human CCL20 polyclonal antibody or control normal rabbit polyclonal IgG. Goat anti-rabbit IgG (H + L) cross-adsorbed secondary antibody with Alexa Fluor 488 dye was used as the secondary antibody. The nucleus was stained with 4′,6-diamino-2-phenylindole (DAPI). Slides were then mounted with Ultra Cruz mounting medium (Santa Cruz Biotechnology, Dallas, TX, USA) and were observed using a D-Eclipse confocal laser scanning microscope (Nikon, Tokyo, Japan).

### 4.7. Statistical Analysis

All data are presented as mean ± standard deviation (SD). The significance of differences between groups was assessed using unpaired 2-tailed student’s *t*-test (2 groups) or 1-way analysis of variance (ANOVA), followed by Tukey’s multiple comparison test (multiple groups) using GraphPad PRISM 7.0 software (GraphPad Software, La Jolla, CA, USA). A *p*-value of < 0.05 was considered statistically significant.

## Figures and Tables

**Figure 1 ijms-21-00434-f001:**
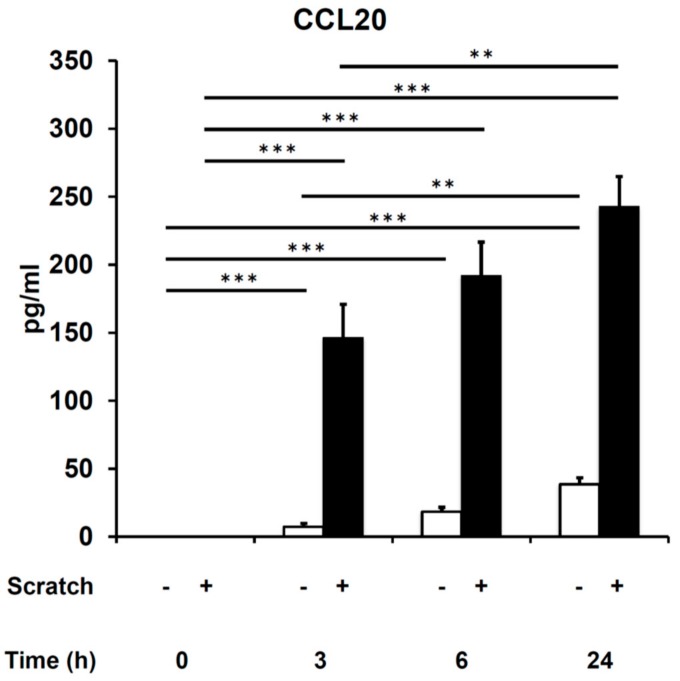
Scratch injury–induced CCL20 production. The production of CCL20 was measured at 3, 6, and 24 h after the initiation of culture in non-scratched control and scratched keratinocyte cultures. Representative data of three independent experiments are shown. ** *p* < 0.01. *** *p* < 0.001.

**Figure 2 ijms-21-00434-f002:**
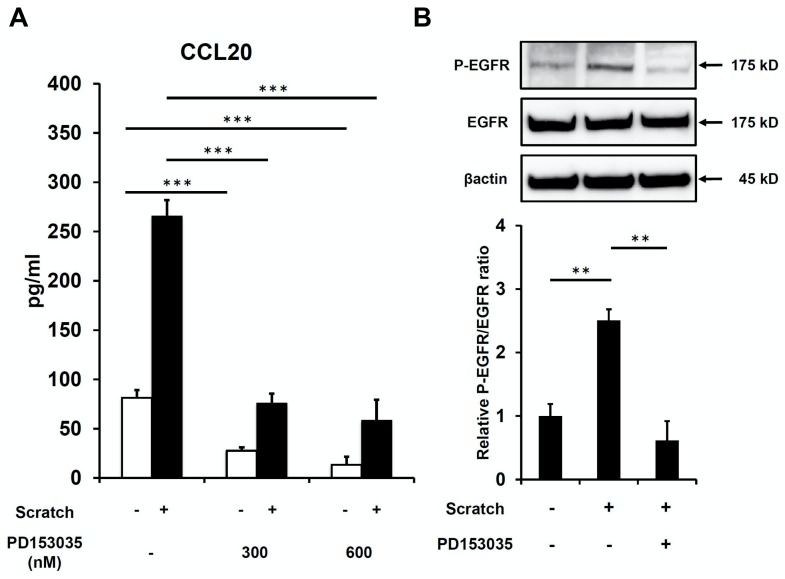
Scratch injury-induced CCL20 production depends on activation of epidermal growth factor receptor (EGFR). (**A**) Scratch injury-induced CCL20 production was measured in the presence or absence of PD153035 (EGFR inhibitor, 300 or 600 nM) at 24 h after scratching. (**B**) The phosphorylation of EGFR (P-EGFR) was measured by western blot analysis at 1 h after scratching. Representative enzyme-linked immunosorbent assay (ELISA) data and Western blot images of three independent experiments are shown. ** *p* < 0.01. *** *p* < 0.001.

**Figure 3 ijms-21-00434-f003:**
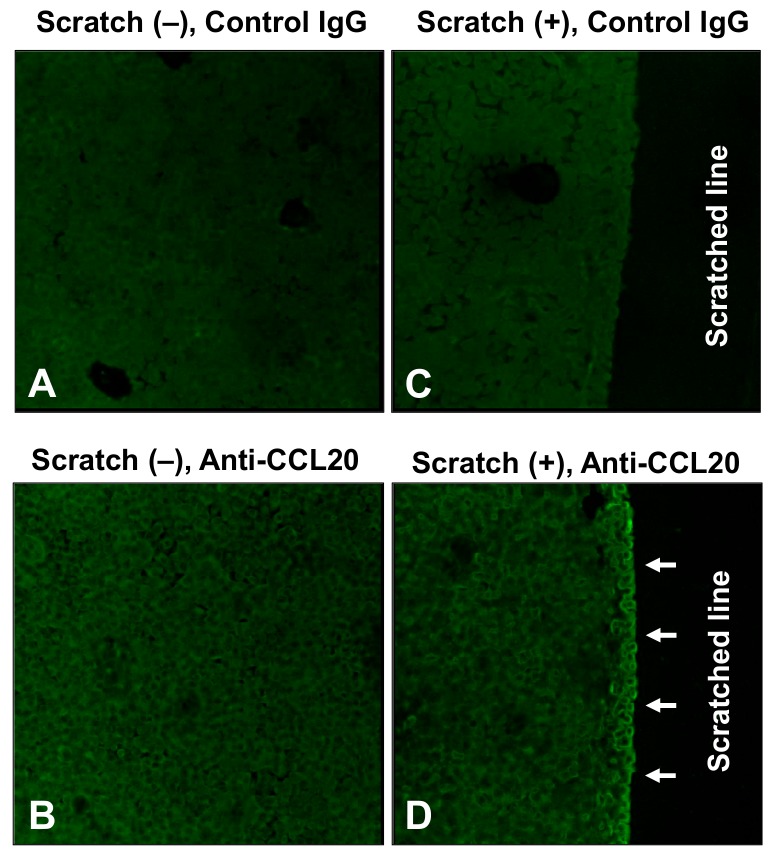
Immunofluorescent visualization of CCL20 at 6 h after scratching. (**A**) Non-scratched control keratinocytes stained with control IgG. (**B**) Non-scratched control keratinocytes stained with anti–CCL20 antibody. (**C**) Scratched keratinocytes stained with control IgG. (**D**) Scratched keratinocytes stained with anti–CCL20 antibody. Original magnification 100×. Representative data of three independent experiments are shown.

**Figure 4 ijms-21-00434-f004:**
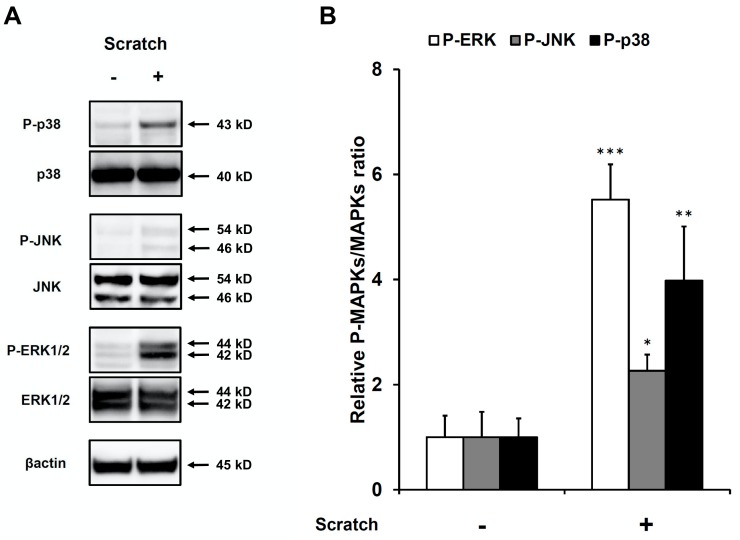
Scratch injury induces phosphorylation of ERK1/2, JNK, and p38 MAPK at 1 h after scratching. Phosphorylation of ERK1/2, JNK, and p38 MAPK was examined by western blotting in non-scratched control and scratched keratinocytes. Representative blot images of three independent experiments (**A**) and the relative expressions of phosphorylated proteins (**B**) are shown. * *p* < 0.05. ** *p* < 0.01. *** *p* < 0.001.

**Figure 5 ijms-21-00434-f005:**
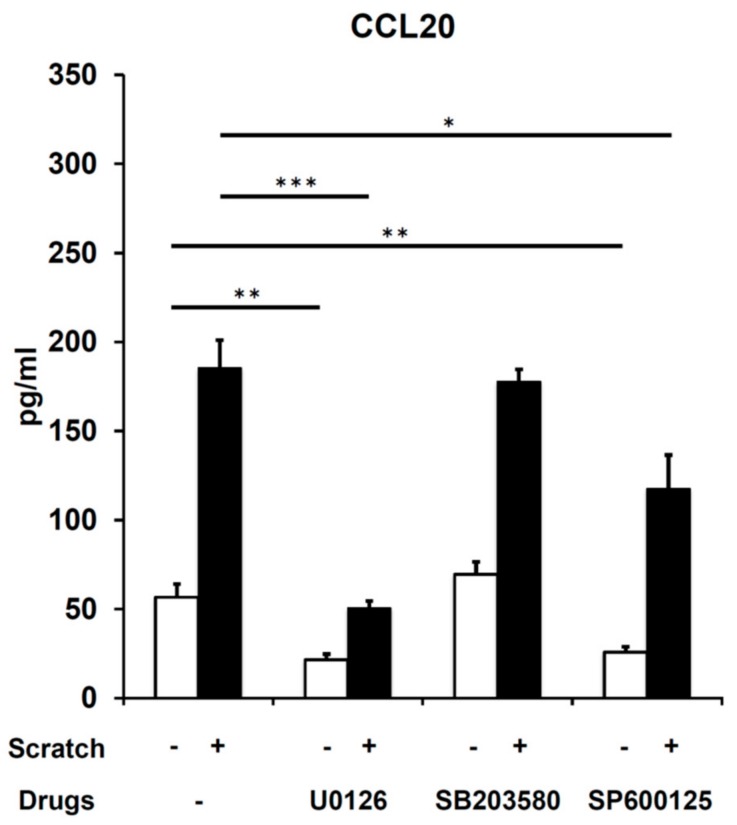
Scratch-induced CCL20 (24 h after scratching) upregulation depends on ERK1/2 and JNK activation. The CCL20 production was measured in nonscratched control and scratched keratinocytes in the presence or absence of U0126 (ERK1/2 inhibitor, 10 μM), SB203580 (p38 MAPK inhibitor, 10 μM), and SP600125 (JNK inhibitor, 10 μM). Representative data of three independent experiments are shown. * *p* < 0.05. ** *p* < 0.01. *** *p* < 0.001.

**Figure 6 ijms-21-00434-f006:**
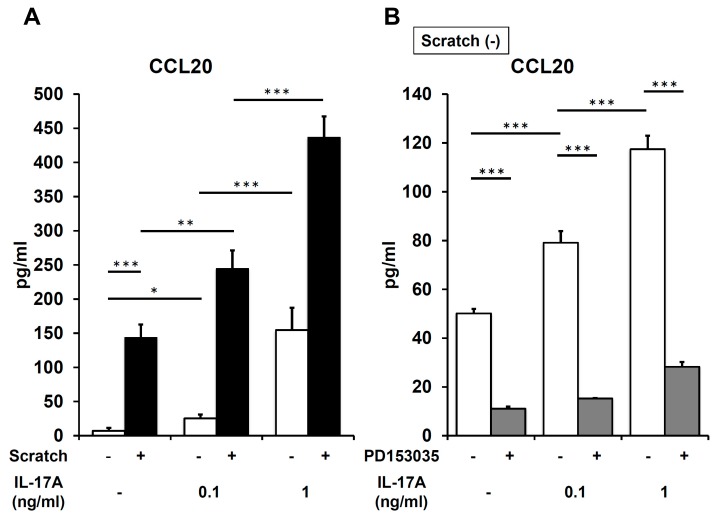
IL-17A augments scratch-induced CCL20 upregulation at 3 h after scratching. (**A**) CCL20 production was measured in non-scratched control and scratched keratinocytes in the presence or absence of IL-17A (0.1 or 1 ng/mL). (**B**) Non-scratched keratinocytes were stimulated with IL-17A (0.1 or 1 ng/mL) with or without PD153035 (EGFR inhibitor, 300 nM). Representative data of three independent experiments are shown. * *p* < 0.05. ** *p* < 0.01. *** *p* < 0.001.

**Figure 7 ijms-21-00434-f007:**
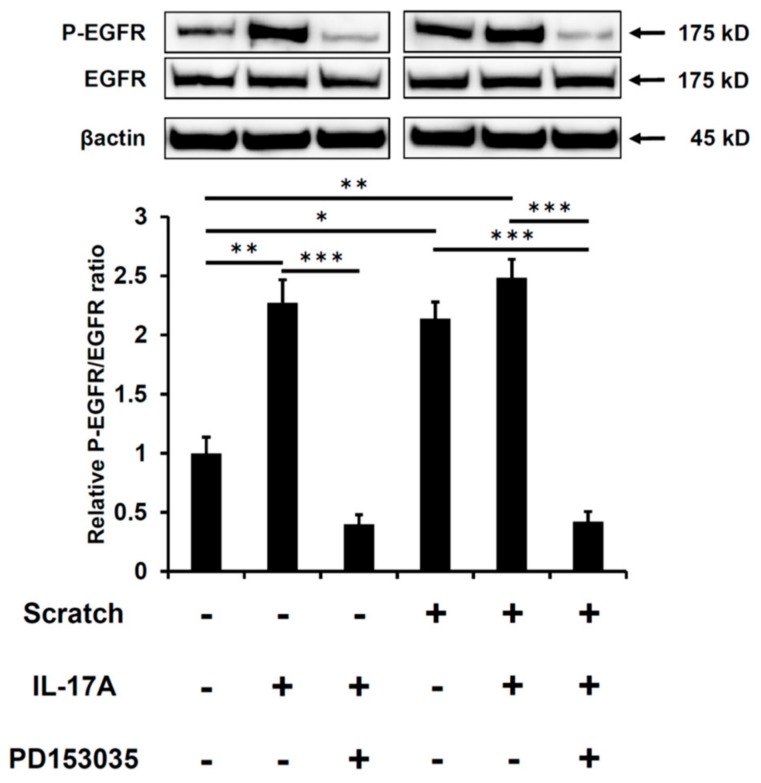
IL-17A upregulates the phosphorylation of EGFR at 3 h after scratching. The phosphorylation of EGFR was measured by western blot in non-scratched control keratinocytes and scratched-keratinocytes in the presence or absence of IL-17A (1 ng/mL) or PD153035 (EGFR inhibitor, 300 nM). Representative blot images of three independent experiments and the relative expressions of phosphorylated proteins are shown. * *p* < 0.05. ** *p* < 0.01. *** *p* < 0.001.

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
