# Peer review of "The EGFR-ERK/JNK-CCL20 Pathway in Scratched Keratinocytes May Underpin Koebnerization in Psoriasis Patients"

_ijms, 2020, doi:10.3390/ijms21020434_

Round 1

Reviewer 1 Report

Introduction - Please add brief additional details on Koebner phenomenon i.e. which diseases exhibit this, what is currently known about the pathogenesis.  Introduction - before the sentence on TNF-alpha and IL17/23A axis, please add that a number of biologic therapies are currently approved for the management of moderate to severe psoriasis. These biologics target specific molecules in the immune system, and they have a favorable safety and efficacy profiles than the traditional systemic agents such as methotrexate and cyclosporine. Reference: https://www.ncbi.nlm.nih.gov/pubmed/30246393  Discussion - please add a sentence on the importance of your findings i.e. does your study underscore the potential role of new therapies such as anti-CCL20 antibody and EGFR/JNK inhibitors in the management of psoriasis. 

Author Response

Reply to the Reviewer 1

→ First of all, we would like to show our appreciation to reviewers for their time to review the manuscript and for their critical and helpful comments. We have revised the manuscript based on the reviewer’s comments and explained as follows.

Introduction - Please add brief additional details on Koebner phenomenon i.e. which diseases exhibit this, what is currently known about the pathogenesis.

→ Thank you very much for your helpful comments. According to your comment, we added the following sentences in the Introduction.

Line 46 to 48

“Although Koebner phenomenon is observed in other skin diseases such as lichen planus and vitiligo, it is particularly associated with psoriasis [5,6]. However, the pathogenetic mechanism of koebnerization is not clear [5,6].”  

Introduction - before the sentence on TNF-alpha and IL17/23A axis, please add that a number of biologic therapies are currently approved for the management of moderate to severe psoriasis. These biologics target specific molecules in the immune system, and they have a favorable safety and efficacy profiles than the traditional systemic agents such as methotrexate and cyclosporine.

Reference: https://www.ncbi.nlm.nih.gov/pubmed/30246393

→ Thank you very much for your helpful comments. According to your comment, we added the following sentences in the Introduction by adding Ref #7.

Line 49 to 52

“A number of biologic therapies are currently approved for the management of moderate to severe psoriasis. These biologics target specific molecules in the immune system, and they have a favorable safety and efficacy profiles than the traditional systemic agents such as methotrexate and cyclosporine [7].”

Discussion - please add a sentence on the importance of your findings i.e. does your study underscore the potential role of new therapies such as anti-CCL20 antibody and EGFR/JNK inhibitors in the management of psoriasis.

→ Thank you very much again for your helpful comments. According to your comment, we added the following sentences in the Discussion.

Line 237 to 239

“Our study underscores the potential role of new therapies such as anti-CCL20 antibody and EGFR/ERK inhibitors in the management of psoriasis.”

.”

Thank you so much again for your helpful comments. We hope the revised article is now suitable for publication in IJMS.

Reviewer 2 Report

Psoriasis is an immune-mediated skin disease characterized by epidermal hyperproliferation. Skin injury triggers or exacerbates psoriatic lesions in a condition named the Koebner phenomenon. The authors have previously demonstrated an upregulated production of CCL20 following scratch injury.  The aim of the present study was to evaluate the possible mechanism of this reaction.

Authors  demonstrated that scratch-induced CCL20 production was mediated by EGFR–extracellular signal–related kinase (ERK) and by the EGFR–c-Jun N-terminal kinase pathway in keratinocytes. IL-17A also upregulated CCL20 production via EGFR activation and further potentiated scratch-induced CCL20 production. The results suggest that epidermal CCL20 production is an integral part in the pathogenesis of psoriasis and koebnerization.

The study was well planned and performed, the figures are informative, and the data justified the conclusions.

The results of the study may contribute to the development and implementation of new methods of psoriasis treatment.  

I have just some minor concerns.

Was the evaluated pathway examined in other skin diseases? Koebner phenomenon is also present in lichen planus. Any data on this disease? Or other? What are the limitations of the study in your opinion? Did the number N=3 is not too small to draw conclusions?

Author Response

Reply to the Reviewer 2

→ First of all, we would like to show our appreciation to reviewers for their time to review the manuscript and for their critical and helpful comments. We have revised the manuscript based on the reviewer’s comments and explained as follows.

Psoriasis is an immune-mediated skin disease characterized by epidermal hyperproliferation. Skin injury triggers or exacerbates psoriatic lesions in a condition named the Koebner phenomenon. The authors have previously demonstrated an upregulated production of CCL20 following scratch injury. The aim of the present study was to evaluate the possible mechanism of this reaction. Authors demonstrated that scratch-induced CCL20 production was mediated by EGFR–extracellular signal–related kinase (ERK) and by the EGFR–c-Jun N-terminal kinase pathway in keratinocytes. IL-17A also upregulated CCL20 production via EGFR activation and further potentiated scratch-induced CCL20 production. The results suggest that epidermal CCL20 production is an integral part in the pathogenesis of psoriasis and koebnerization. The study was well planned and performed, the figures are informative, and the data justified the conclusions. The results of the study may contribute to the development and implementation of new methods of psoriasis treatment.

→ Thank you very much for your encouraging comment.

I have just some minor concerns. Was the evaluated pathway examined in other skin diseases? Koebner phenomenon is also present in lichen planus. Any data on this disease? Or other?

→ Thank you very much for your helpful comment. Despite a few reports, increased levels of CCL20/CCR6 has been reported. According to your comment, we added the following sentences in the Discussion by adding ref. #57.

Line 228 to 229

“In parallel, increased expression of CCL20 and CCR6 has been demonstrated in lichen planus which also manifests koebnerization [57].”

What are the limitations of the study in your opinion? Did the number N=3 is not too small to draw conclusions?

→ Thank you very much for your helpful comment. According to your comment, we added the following sentences in the revised Discussion.

Line 229 to 234

“There are several limitations in this study. We only examined the effects of scratch injury on the monolayer keratinocyte culture system. Further studies are warranted using 3-dimensional skin equivalent, ex vivo human skin or in vivo mouse models. Fibroblasts may affect the scratch-induced, keratinocyte-derived CCL20 expression in coculture models. More extensive studies may be necessary to draw a concrete conclusion.”

Thank you so much again for your helpful comments. We hope the revised article is now suitable for publication in IJMS.
